# Exploratory Study of the Relationship between an Oral Fungal Swab Test and Patient Blood Test Data

**DOI:** 10.3390/microorganisms11122887

**Published:** 2023-11-29

**Authors:** Tatsuki Itagaki, Ken-ichiro Sakata, Akira Hasebe, Yoshimasa Kitagawa

**Affiliations:** 1Oral Diagnosis and Medicine, Faculty of Dental Medicine, Graduate School of Dental Medicine, Hokkaido University, Kita-13 Nishi-7, Kita-ku, Sapporo 060-8586, Japan; titagaki@den.hokudai.ac.jp (T.I.); sakata-0303@den.hokudai.ac.jp (K.-i.S.); ykitagaw@den.hokudai.ac.jp (Y.K.); 2Oral Molecular Microbiology, Faculty of Dental Medicine, Graduate School of Dental Medicine, Hokkaido University, Kita-13 Nishi-7, Kita-ku, Sapporo 060-8586, Japan

**Keywords:** oral *Candida*, *Nakaseomyces glabrata*, serum albumin, aging

## Abstract

Our understanding of the relationship between oral *Candida* and systemic conditions has significantly increased recently, which this study aims to extend further by investigating the risks of oral candidiasis. A total of 314 patients were involved in this study and underwent an oral swab test at the Department of Oral Medicine, Hokkaido University Hospital, between January and December 2021. Data were collected on age, sex, white and red blood cell counts, Hb, total protein, vitamin B12, as well as serum albumin, iron, copper, and zinc levels. The clinical fungus samples were swabbed to identify those with *Candida* species using a MALDI Biotyper, then applied analysis of covariance and multivariant logistic regression analysis. It was possible to assess the oral swab test results without considering the difference between sex (*p* = 0.946). The oral swab test results were associated with aging (odds ratio: 1.03) and serum albumin levels (odds ratio: 0.32). In summary, the results of our study suggest a relationship between aging and oral candidiasis and offer in-depth insights into how to prevent or treat oral candidiasis onset.

## 1. Introduction

An estimated 30–60% of healthy adults carry *Candida* species within their oral cavities [1].

*Candida* detection in the oral cavity has recently increased in the elderly [2]. Oral candidiasis is an opportunistic infection. *Candida* infection is usually treated with antifungal drugs, although fewer types of these drugs are available compared to antibacterial drugs. Some cases of azole- or echinocandin-resistant *C. albicans* have been observed worldwide [2,3]. Fungal drug resistance reportedly represents a treatment hurdle [2,3]. Oral candidiasis is an opportunistic infection with several potential risk factors, such as age, dentures, immunosuppression, dysbiosis, dry mouth, malnutrition, endocrine abnormalities, and anemia [4]. Moreover, the number of dental caries and *Candida* colony-forming units in the saliva reportedly correlate [5]. However, many papers have reported analytical results that divided continuous data into arbitrary segments, which reduced the amount of information contained in the continuous data [6,7,8,9,10,11]. As a result, information on the continuity is lost during the data segmentation process and becomes discrete data, making it difficult to estimate the results in detail [6,7,8,9,10,11]. This process involves converting the data into discrete data, which reduces their reliability. If age is divided into two categories based on the age of 65 years, individuals under 65 years are considered young and those over 65 years are considered elderly. It is possible to determine if a person is over 65 years old, but not someone’s exact age. This is because the information contained in the data is discarded by segmenting it, which reduces the reliability of the analysis. In this study, continuous data are analyzed as continuous values to avoid loss of the information contained in continuous data. In this regard, using analysis of covariance helps to reduce data errors and to perform highly accurate analyses by assuming that the regression lines for the two groups are parallel (Figure 1). Therefore, the aim of this study is to find a statistical relationship between the physical and biochemical characteristics of adult people and their fungus carrier status.

The results of this study will offer in-depth insights into how to prevent or treat oral candidiasis onset and elucidate the underlying mechanisms of opportunistic infection pathology by deciphering the relationships between certain risk factors using multivariate analysis.

## 2. Materials and Methods

### 2.1. Patients and Data Collection

This study entailed a retrospective search for related factors in the examination data of patients who visited the Department of Oral Medicine, Hokkaido University Hospital, as their first visit from January to December 2021.

Data were collected on age, sex, white and red blood cell (WBC and RBC, respectively) counts, Hb, total protein, vitamin B12, as well as serum albumin, iron, copper, and zinc levels. Clinical fungus samples were swabbed (Eiken Chemical Co., Ltd., Tokyo, Japan) and the specimens were promptly transported to the laboratory at our hospital. Mass spectrometry with a MALDI Biotyper ^®^ (Bruker Daltonics, Billerica, MA, USA) was used to identify the *Candida* species. The MALDI Biotyper analysis was performed according to the manufacturer’s instructions.

This retrospective study was conducted with the approval of the Hokkaido University Hospital Independent Clinical Research Review Committee (Approval No. 202-0049). All the study procedures were performed in accordance with the principles of the Declaration of Helsinki. This article does not disclose identifiable information about any of the participants in any form. Hence, no written consent for publication is applicable in this case.

### 2.2. Statistical Analysis

The prevalence was calculated, and its 95% confidence interval (CI) was estimated using the Agresti method. The relationships between variables were analyzed using an analysis of covariance (ANCOVA) and multivariate logistic regression (the method of maximum likelihood). The relationship between sex and the co-infection of *C. albicans* and *Nakaseomyces glabrata* (*syn. Candida glabrata*) were examined by Cramer’s coefficient of association. The significance level was set at *p* < 0.05 for all analyses.

The data were analyzed using Excel (Microsoft^®^ Excel^®^ for Microsoft 365 MSO (version 2306, build 16.0.16529.20164, 64 bit) and R version 4.0.3 (2020-10-10) (Copyright © 2020, The R Foundation for Statistical Computing).

## 3. Results

The dataset included 314 participants and the *Candida* species positive rate was 28.0% (CI: 23.1–33.3%).

As summarized in Table 1, the age, serum albumin, and hemoglobin levels as well as the RBC count were imbalanced between the two groups, while the WBC count was only slightly imbalanced between the two groups. Several *Candida* species were detected in the oral cavity (Table 2).

The results of the analysis of covariance indicate that there is almost no interaction between sex and age (*p* = 0.946). The effect of sex adjusted for age was, therefore, considered small (*p* = 0.403), while that of the opposite was considered large (*p* < 0.05) (Table 3 and Figure 1).

Table 4 presents the results of the univariate logistic regression analysis, although these results are less important as they contain bias.

Subsequently, a multivariate analysis was performed, adjusting for factors that were imbalanced between the two groups (Table 5). Regarding the anemia evaluation index, Hb was selected in consideration of multicollinearity. However, Hb was strongly associated with sex, and anemia and hypoalbuminemia correlated with each other. Hence, Hb could be excluded from the factors. The results of the multivariate analysis, excluding Hb, were summarized and it was concluded that the difference in WBC was negligible (Table 6). The best logistic regression model for our study is shown in Table 7, with odds ratios of 1.03 and 0.32 for aging and serum albumin, respectively. The selected variables were the same results as those of the stepwise reduction using *p*-values.

Finally, Table 8 presents the *C. albicans* and *N. glabrata* co-infection-related patient data. The prevalence of the *N. glabrata* carriers among the *Candida* species carriers was 0.1 (0.05–0.19). As the sample size was small and an imbalance could be detected between the groups, no comparisons were possible between them. Moreover, a high amount of missing blood data was present in this case; therefore, these comparisons are unreliable. Thus, a less reliable univariate analysis for sex was performed. Cramer’s coefficient of association between sex and *N. glabrata* was 0.08 (0.005–0.30); therefore, no associations were observed.

## 4. Discussion

A cross-sectional study is an appropriate study design for prevalence research. Causal relationships cannot be investigated by cross-sectional studies. It is possible to predict the relationship between events by approximating them by using random variables. It is generally beneficial to use approximate predictions in exploratory studies. ANCOVA is a type of ANOVA that controls the linear effect of covariate variables by using regression analysis. Our analysis of covariance indicated no sex-related differences in the case of *Candida* infections. Thus, this study concluded that the consideration of the difference in the screening rates of *Candida* infection occurs equally regardless of sex. Since the elements were balanced in both groups, their effect was insignificant. Moreover, the WBC level might not be associated with *Candida* positivity, based on the odds ratios in our multivariate analysis.

A previous study compared age in 10-year categories [6], meaning that differences of 1–9 years were considered in the same category. As a result from data segmentation, differences could be observed for participants in their eighties and above [6]. Moreover, our study showed that the patients in whom *Candida* was detected were 6 years older on average than those in whom *Candida* was detected, with an odds ratio of 1.03 (positive probability = exp(−2.88 + 0.03 × age)). Our dataset indicated a *Candida*-positive sensitivity and specificity of 0.66 and 0.58, respectively (AUC: 0.63 (0.56–0.7)) beyond the age of 70 years. These data suggest an over 50% chance of positivity at the ages of 70–80 years and above. The diagnostic accuracy of this discriminant is affected by factors such as prevalence. Considering prevalence, the positive and negative predictive values were 0.4–0.6 and 0.8–0.6, respectively, indicating that beyond 70 years of age, the chances of *Candida* positivity are 40–60%. Aging contributes to a positive oral *Candida* test. The odds ratios for age and oral swab test results were consistent with those of 423 individuals, which were collected at different times (Appendix A).

A high prevalence of oral *Candida* infection has been reported in patients with iron deficiency anemia [8]. This relationship was not clear in this study. Chronic inflammations reduce serum albumin levels and the RBC count. Moreover, malnutrition reduces serum albumin levels as well as RBC and hemoglobin counts. Elderly people are prone to anemia and malnutrition, both leading to infectious diseases. Nutritional intake can be intervened by a dentist. Therefore, dentists must treat patients so that they can chew properly. This is to increase the efficiency of nutrient absorption, which will improve serum albumin levels.

Previous preliminary results are available on *C. albicans* and *N. glabrata* co-infection. The prevalence of *N. glabrata* carriers among *Candida* species carriers was 0.18 (0.12–0.25), and *N. glabrata* could not be associated with sex (*V*: 0.003–0.33) [9]. According to our data, *N. glabrata* cannot be associated with sex either (*V*: 0.005–0.30). Two independent studies yielded similar results related to prevalence, age, and sex. Therefore, the results of these studies are small but not unreliable. *N. glabrata* might be associated with denture use (*V*: 0.13–0.47, *p* = 0.0007) [9]. Whether the patient used dentures could not be determined based on the medical record information consulted in this study. Certain studies focused on the relationship between dentures and *Candida* abundance or denture size and *Candida* colonization frequency [10,11]. In general, the probability of denture use increases with age. Therefore, age might be a confounding factor in *C. albicans* and *N. glabrata* co-infection [9]. When *N. glabrata* is the source of infection, it is likely associated with dentures [9]. Therefore, the source of infection could be potentially eliminated by performing an intraoral check. *N. glabrata* is naturally resistant to azole antifungal agents [12]. Denture cleaning and oral care by a dental professional could thus represent an effective treatment.

These results are consistent with existing research that revealed that older age is a risk factor [6,7,8,9,10,11]. The meaning of “age extremes” was confirmed by this study [1]. Additionally, there is disagreement regarding the effect of sex [6,7,8,9,10,11], but this study concluded that it was unrelated. Trace elements had a negligible effect compared to others and might only become a risk factor in special circumstances [8]. The odds ratio is a statistic that shows association. Correlation and causation are different concepts. Sometimes, things appear to be related even if they are not. Although the influence of each factor was estimated by the odds ratios, the magnitude of the influence of unmeasured factors is unknown. Multivariate analysis is a relative evaluation of variables. It is also possible that the effect of aging is smaller than the effect of the unmeasured factors. A previous study demonstrated that *Candida* mannan concentrations were higher in subjects older than 80 years of age with a higher number of either untreated or prosthetic teeth, a lower salivary pH, and a reduced RBC count [6]. However, denture use, current tooth number, and oral hygiene state were not measured as confounding factors, since these data could not be obtained from the medical records. Although these are the limits of the study design and prediction based on approximation, this study presented a more impactful evaluation than the existing results. Future studies should focus on elucidating the relationship between serum albumin levels and diseases through cohort studies. Furthermore, it is necessary to examine the causal relationship between the use of dentures and the onset of oral candidiasis, adjusting for age. Studies on *C. albicans* and *N. glabrata* co-infection can calculate sample size from the prevalences. Since the sample size is 390 (>1/0.26 × 1/0.1 × 10) people per one variant, research using the Mantel–Haenszel test is preferable. The results in Table 8 of this study can be used for meta-analysis studies that apply the Mantel–Haenszel test. This study can therefore help to draw new specific hypotheses.

## 5. Conclusions

In this study, the detection rate of *Candida* in the oral cavity and the prevalence of *N. glabrata* carriers among *Candida* species carriers were approximately 30% and 0.05–0.25, respectively. Sex and trace elements are unrelated to positive *Candida* tests. However, oral candidiasis development is complex. Serum albumin levels and aging were associated with the oral *Candida* test results. The older the patient, the more likely they are of testing positive for *Candida*. Therefore, paying attention to fungal infections is imperative, especially beyond the age of seventy years. Further research is needed to clarify these complex relationships.

## Figures and Tables

**Figure 1 microorganisms-11-02887-f001:**
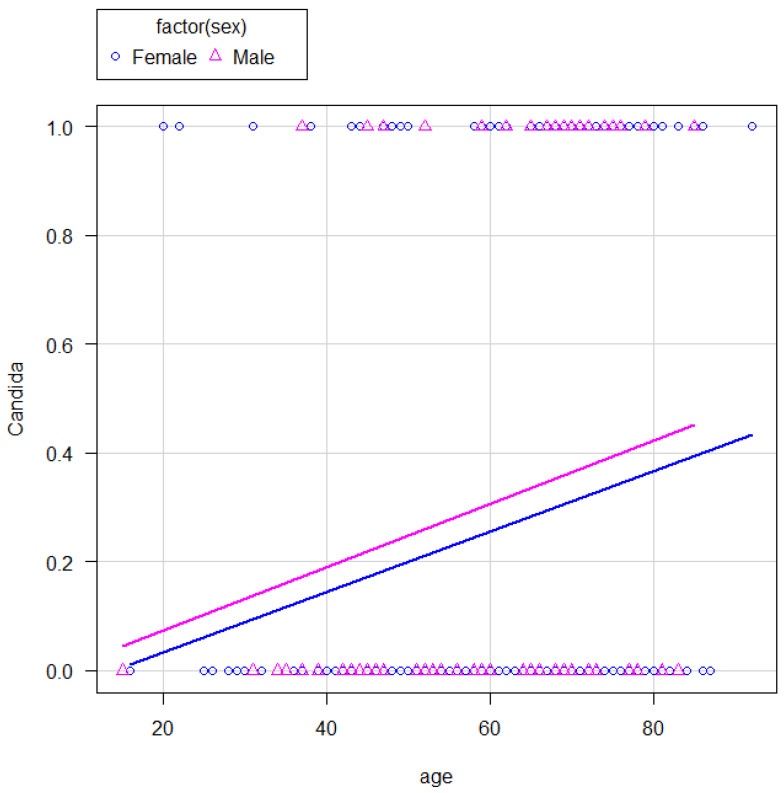
ANCOVA results. The regression lines for each sex were parallel (*p* = 0.946). Based on the regression lines, almost no sex-related effect could be detected (*p* = 0.403). Older people were more likely to test positive (*p* < 0.05).

**Table 1 microorganisms-11-02887-t001:** Patient demographic data 1.

Factor	Group	Negative	Positive	SMD	*n*
*n*	-	226	88	-	314
Sex (%)	Female	181 (80.1)	68 (77.3)	0.069	249
	Male	45 (19.9)	20 (22.7)		65
Age (mean (SD)) year		60.96 (14.98)	67.12 (14.43)	0.419	314
Alb (mean (SD)) g/dL		4.30 (0.29)	4.18 (0.34)	0.408	255
Cu (mean (SD)) μg/dL		112.46 (18.98)	112.32 (18.01)	0.007	234
Fe (mean (SD)) μg/dL		87.98 (29.51)	85.77 (30.88)	0.073	232
Hb (mean (SD)) g/dL		13.62 (1.57)	13.09 (1.23)	0.376	256
RBC (mean (SD)) × 10^6^/μL		4.47 (0.50)	4.27 (0.44)	0.438	256
TP (mean (SD)) g/dL		7.08 (0.36)	7.03 (0.41)	0.122	255
VB12 (mean (SD)) pg/dL		722.06 (592.73)	723.44 (707.61)	0.002	228
WBC (mean (SD)) × 10^4^/μL		5.77 (1.66)	6.08 (1.97)	0.169	256
Zn (mean (SD)) μg/dL		78.63 (13.49)	77.48 (10.82)	0.094	237

SMD: Standardized mean difference.

**Table 2 microorganisms-11-02887-t002:** Candida species and *Nakaseomyces glabrata* (syn. Candida glabrata) case counts.

*Candida* Species	Counts
*C. albicans*	67
*N. glabrata*	5
*Meyerozyma guilliermondii* (*syn. Candida guilliermondii*)	1
*C. tropicalis*	0
*C. parapsilosis*	2
*C. albicans* + *N. glabrata*	8
*C. albicans* + *M. guilliermondii*	3
*C. tropicalis* + *N. glabrata*	1
*C. albicans* + *C. tropicalis*	1

**Table 3 microorganisms-11-02887-t003:** ANCOVA results.

ANOVA Table (Type III tests)	*p*-Value
Interaction	0.946
Factor (sex)	0.403
Age	0.000854

**Table 4 microorganisms-11-02887-t004:** Results of univariate analysis 1.

Logistic Regression Analysis	Odds Ratio	95% CI	*p*-Value	*n*
age	1.03	(1.01, 1.05)	0.0013	314
Alb	0.25	(0.09, 0.66)	0.0052	255
Cu	1.00	(0.98, 1.02)	0.96	234
Fe	1.00	(0.99, 1.01)	0.64	232
Hb	0.79	(0.65, 0.97)	0.021	256
RBC	0.41	(0.22, 0.76)	0.0051	256
TP	0.72	(0.33, 1.60)	0.40	255
VB12	1.00	(1.00, 1.00)	0.99	228
WBC	1.10	(0.94, 1.30)	0.23	256
Zn	0.99	(0.97, 1.02)	0.57	237

CI: Confidence interval.

**Table 5 microorganisms-11-02887-t005:** Results of the multivariate analysis 1.

Logistic Regression Analysis	Odds Ratio	95% CI	*p*-Value	*VIF*
Analysis of Deviance	-	-	0.0011	-
Age	1.03	(1.00, 1.05)	0.02	1.03
Alb	0.44	(0.15, 1.32)	0.14	1.19
WBC	1.14	(0.96, 1.35)	0.14	1.05
Hb	0.85	(0.68, 1.06)	0.15	1.22

*VIF*: Variance inflation factor.

**Table 6 microorganisms-11-02887-t006:** Result of the multivariate analysis 2.

Logistic Regression Analysis	Odds Ratio	95% CI	*p*-Value	*VIF*
Analysis of Deviance	-	-	0.0010	-
Age	1.03	(1.01, 1.05)	0.01	1.03
Alb	0.33	(0.12, 0.90)	0.03	1.03
WBC	1.11	(0.94, 1.31)	0.24	1.00

**Table 7 microorganisms-11-02887-t007:** Results of the multivariate analysis 3.

Logistic Regression Analysis	Odds Ratio	95% CI	*p*-Value	*VIF*
Analysis of Deviance	-	-	0.00058	-
Age	1.03	(1.01, 1.05)	0.01	1.03
Alb	0.32	(0.12, 0.88)	0.03	1.03

**Table 8 microorganisms-11-02887-t008:** Demographic data of patients 2.

Factor	Group	*Candida albicans*	*C. albicans + N. glabrata*	SMD
*n*	-	72	8	-
Sex (%)	Female	55 (76.4)	7 (87.5)	0.29
	Male	17 (23.6)	1 (12.5)	
Age (mean (SD)) year		65.5 (15.0)	75.5(9.6)	0.79
Alb (mean (SD)) g/dL		4.21 (0.33)	4.04 (0.23)	0.58
Cu (mean (SD)) μg/dL		111.4 (18.8)	117.3 (15.0)	0.35
Fe (mean (SD)) μg/dL		86.2 (33.5)	92.1 (17.8)	0.22
Hb (mean (SD)) g/dL		13.2 (1.22)	12.5 (0.75)	0.72
RBC (mean (SD)) × 10^6^/μL		4.3 (0.44)	4.1 (0.30)	0.63
TP (mean (SD)) g/dL		7.1 (0.37)	7.0 (0.37)	0.09
VB12 (mean (SD)) pg/dL		741.7 (775.5)	557.43 (280.0)	0.32
WBC (mean (SD)) × 10^4^/μL		6.0 (1.7)	6.9 (3.5)	0.32
Zn (mean (SD)) μg/dL		76.7 (11.0)	78.6 (7.3)	0.21

## Data Availability

The data that support the findings of this study are available from the corresponding author upon reasonable request.

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
