# Peer review of "Exploratory Study of the Relationship between an Oral Fungal Swab Test and Patient Blood Test Data"

_microorganisms, 2023, doi:10.3390/microorganisms11122887_

Round 1

Reviewer 1 Report

Comments and Suggestions for Authors

The introduction should be expanded providing more details and references updated

The discussion should compare the findings with more data retrieved from the literature.

Moreover strong and weak point of this paper should be discussed.

Comments on the Quality of English Language

English language is fine

Author Response

Dear Reviewer 1,

Thank you very much for reviewing our manuscript and offering valuable advice.

We have addressed your comments with point-by-point responses, and revised the manuscript accordingly. To make our present study easier to understand, we modified our manuscript according to your suggestion.

The introduction should be expanded providing more details and references updated

Answer

Thank you so much for your suggestion. The introduction was ambiguous and difficult-to-read. We decided to explain its ambiguous point with a concrete example. And to help understand We added statistical interpretation in discussion. This is because there are various schools of statistics.

Therefore, we will explain general scientific research in detail, so you can understand the content of our research.

First, we must clarify the research topic by posing a problem.

Next, we will conduct exploratory research, such as surveys and observations, and scientifically interpret and evaluate the results. This study is an observational study, which means that it is an observed fact. Observed facts include events that occur by chance, and may have no basis or mechanism. However, at the stage of formulating a plan for exploratory research, a certain amount of theory and qualitative hypotheses should be assumed. In order to select evaluation items and evaluation indicators (summary values or representative values) to be used in verification-type research, we try to observe as many candidate items as possible. Descriptive statistical methods will be used to summarize the data. The objective facts obtained at this stage are mainly qualitative, and are interpreted and evaluated scientifically. We collected as many items as possible from past studies that seemed to be related, but we were unable to collect items such as oral hygiene, diets, diabetes and lifestyles, etc. as they were rarely recorded in medical records. Therefore, there is no explanation for these missing values.

This study aims to formulate a new hypothesis. From here on, our research will proceed beyond this point, so although it is not relevant to this topic, we will explain it to help your understanding. Build scientific theories and models based on the results obtained through exploratory research, and set quantitative hypotheses derived from them. At this time, the hypothesis should be such that the validity and validity of the theory or model can be evaluated.

We plan, draft, and implement verification-type research, such as experiments and tests, and scientifically interpret and evaluate the results. The main purpose of verification-type research is to verify a hypothesis and evaluate the validity and validity of a theory, and confirmation tests and main experiments correspond to this. At the stage of planning a verification-type research, determine the evaluation items, evaluation indicators, statistical methods, significance level, reliability coefficient, etc. for testing the hypothesis, and calculate the required number of samples. Use inferential statistical methods to summarize the data. The objective facts obtained at this stage are mainly quantitative, and are interpreted and evaluated scientifically.

Finally, the hypothesis is confirmed or revised based on the results obtained in the test-type research, and the scientific theory is confirmed or revised.

This allows new questions to be raised, theories to be revised, etc., and feedback to the appropriate research stage.

We will also explain the research design and what can be done.

Research designs can be broadly divided into observational studies and experimental studies. We will explain on observational studies. Observational studies can be further classified into cross-sectional studies, prospective studies, and retrospective studies.

Cross-sectional research is cross-sectional in terms of the direction of causality, meaning that both cause and effect are observed without being fixed. We specify the total number of cases, examine the presence or absence of the cause and the presence or absence of the effect at a certain point in time, and observe both. However, since it is not possible to verify causal relationships, cause and effect are only assumptions or convenient. Instead, both causes and effects can be thought of as random variables, reflecting the natural association between them. Therefore, correlation analysis methods can be applied strictly, and the correlation coefficient has accurate meaning. Cramer's association coefficient corresponds to the correlation coefficient of classification data. This value is zero when there is no relationship between the presence or absence of the risk factor and the presence or absence of the disease, and one when there is a complete relationship. Odds ratios can also be defined. Since both the numerator and denominator of the odds ratio value are random variables, there is an error, making the value less reliable. And in some cases, it may not be possible to calculate. Odds ratios are suitable for use as an indicator of association when the relationship between a risk factor and a disease is close to an exponential relationship (when the prevalence of the disease is small). However, this value has unlimited upper and lower limits and may not be calculated in some cases.

Prospective research is a research method that observes data from a certain point into the future. Therefore, it is mainly used for verification-type research, and cohort studies are a typical example. This is research that fixes the cause and observes the effect. We specify the number of cases without a cause and the number of cases with a cause, check the result of those groups over time, and observe each result. Therefore, causal relationships can be verified. Since only the result is a random variable, regression analysis methods can be applied strictly, and the regression line has accurate meaning.

A typical example of a retrospective study is a case-control study. We will not explain this study design here.

We should probably add these as you say, but these are obvious, so we omitted them like another epidemiological research.

The discussion should compare the findings with more data retrieved from the literature.

Answer

Comparisons using the same scale are preferable. However, even if we don't compare, in detail, the results were generally consistent with existing rough results. Since the comparison data retrieved from the literature are at the lower of the scale, a good comparison cannot be made. There is no point in comparing the ratio scale (accurate age) with extreme ages such as those over 65 or 80 years old( nominal scale or ordinal scale). Moreover, the study we referred to is a preliminary study in which Candida identification is based on colony color. Rather than comparing the results, it is more profitable to accurately estimate the true values using interval estimations, as we did. And the references were checked and then rewritten.

We can provide another example.

X1 : = (Marked improvement, Moderate improvement, Mild improvement, No change, Worsening) (Vector representation of variables representing order.)

X2 : = (8, 12, 45, 35, 0) (A vector representing observations for Group 1.)

X3 : = (0, 25, 25, 35, 15) (A vector representing observations for Group 2.)

These vectors represent ordinal data. When performing a U-test on the overall degree of improvement, Group 1 showed better improvement (p = 0.04). In this case, the results are reliable.

By setting a cutoff point based on a certain ranking, this data can be segmented.

When classifying mild improvement or better as an improvement, the improvement rate for Group 1 is 65% and that for Group 2 is 50% (chi-square test p value is p = 0.0452). Similarly, when a moderate or higher improvement is defined as an improvement, the improvement rates are reversed between Groups 1 and 2, with Group 1 having a 20% improvement rate and Group 2 having a 25% improvement rate. These results are less reliable and rougher.

It is our belief that comparisons to research results that show such trends should not be made.

Moreover strong and weak point of this paper should be discussed.

Answer

By summarizing the results, we made clear the strong and weak points of this paper. Accurate prevalence surveys were conducted, and various variables were summarized in tables for use in meta-analyses. Added sample size calculations and briefly described meta-analysis methods.

Again, thank you for giving us the opportunity to strengthen our manuscript with your valuable comments. We have worked hard to incorporate your feedback and hope that these revisions persuade you to accept our submission.

Reviewer 2 Report

Comments and Suggestions for Authors

The authors explored the relationship between oral fungal swab test and patient blood test data. However, I can't see the rationale of the current study. The authors must explain the rationale of the current study and the possible mechanism behind it properly. 

The other oral fungal infection related patient information is not explored, such as oral hygiene, diets, diabetics and lifestyles, etc.

Author Response

Dear Reviewer 2,

Thank you very much for reviewing our manuscript.

We have addressed your comments with point-by-point responses, and revised the manuscript accordingly.

Our research content is not related to your comment except our study limitation. Your comment concerns additional research that should be carried out after our current research. It is a mistake to demand an explanation for something that could not be done due to research limitations. Therefore, we will explain general scientific research in detail, so you can understand the content of our research.

First, we must clarify the research topic by posing a problem.

Next, we will conduct exploratory research, such as surveys and observations, and scientifically interpret and evaluate the results. This study is an observational study, which means that it is an observed fact. Observed facts include events that occur by chance, and may have no basis or mechanism. However, at the stage of formulating a plan for exploratory research, a certain amount of theory and qualitative hypotheses should be assumed. In order to select evaluation items and evaluation indicators (summary values or representative values) to be used in verification-type research, we try to observe as many candidate items as possible. Descriptive statistical methods will be used to summarize the data. The objective facts obtained at this stage are mainly qualitative, and are interpreted and evaluated scientifically. We collected as many items as possible from past studies that seemed to be related, but we were unable to collect items such as oral hygiene, diets, diabetes and lifestyles, etc. as they were rarely recorded in medical records. Therefore, there is no explanation for these missing values.

This study aims to formulate a new hypothesis. From here on, our research will proceed beyond this point, so although it is not relevant to this topic, we will explain it to help your understanding. Build scientific theories and models based on the results obtained through exploratory research, and set quantitative hypotheses derived from them. At this time, the hypothesis should be such that the validity and validity of the theory or model can be evaluated.

We plan, draft, and implement verification-type research, such as experiments and tests, and scientifically interpret and evaluate the results. The main purpose of verification-type research is to verify a hypothesis and evaluate the validity and validity of a theory, and confirmation tests and main experiments correspond to this. At the stage of planning verification-type research, determine the evaluation items, evaluation indicators, statistical methods, significance level, reliability coefficient, etc. for testing the hypothesis, and calculate the required number of samples. Use inferential statistical methods to summarize the data. The objective facts obtained at this stage are mainly quantitative, and are interpreted and evaluated scientifically.

Finally, the hypothesis is confirmed or revised based on the results obtained in the test-type research, and the scientific theory is confirmed or revised.

This allows new questions to be raised, theories to be revised, etc., and feedback to the appropriate research stage.

We will also explain the research design and what can be done.

Research designs can be broadly divided into observational studies and experimental studies. We will explain on observational studies. Observational studies can be further classified into cross-sectional studies, prospective studies, and retrospective studies.

Cross-sectional research is cross-sectional in terms of the direction of causality, meaning that both cause and effect are observed without being fixed. We specify the total number of cases, examine the presence or absence of the cause and the presence or absence of the effect at a certain point in time, and observe both. However, since it is not possible to verify causal relationships, cause and effect are only assumptions or convenient. Instead, both causes and effects can be thought of as random variables, reflecting the natural association between them. Therefore, correlation analysis methods can be applied strictly, and the correlation coefficient has accurate meaning. Cramer's association coefficient corresponds to the correlation coefficient of classification data. This value is zero when there is no relationship between the presence or absence of the risk factor and the presence or absence of the disease, and one when there is a complete relationship. Odds ratios can also be defined. Since both the numerator and denominator of the odds ratio value are random variables, there is an error, making the value less reliable. And in some cases, it may not be possible to calculate. Odds ratios are suitable for use as an indicator of association when the relationship between a risk factor and a disease is close to an exponential relationship (when the prevalence of the disease is small). However, this value has unlimited upper and lower limits and may not be calculated in some cases.

Prospective research is a research method that observes data from a certain point into the future. Therefore, it is mainly used for verification-type research, and cohort studies are a typical example. This is research that fixes the cause and observes the effect. We specify the number of cases without a cause and the number of cases with a cause, check the result of those groups over time, and observe each result. Therefore, causal relationships can be verified. Since only the result is a random variable, regression analysis methods can be applied strictly, and the regression line has accurate meaning.

A typical example of a retrospective study is a case-control study. We will not explain this study design here.

You seem to have misunderstood about our research, so we explain this study again.

We explored the relationship between oral fungal swab test and patient blood test data. We authored this paper for clinical researchers to help make a new specific hypothesis. This study is an exploratory study to find a theoretical basis. Therefore, the theoretical basis does not yet exist and cannot be written. We observed that age-related changes and hypoalbuminemia were associated with Candida test results. The rationale of the current study and the possible mechanism behind it that you require requires verification-type research. For this reason, it does not match the purpose of this study and should be sought in another study. If our research requires something that cannot be done, then the research design is certainly inappropriate. However, since this is different from our purpose, raising this as an issue is misunderstanding the purpose of this research. It would be a mistake to ask for these explanations, since the purpose of this study is to formulate hypotheses regarding the rationale for the current study and possible mechanisms behind it. We will respond to these requests after conducting verification-type tests in the future.

The authors explored the relationship between oral fungal swab test and patient blood test data. However, I can't see the rationale of the current study. The authors must explain the rationale of the current study and the possible mechanism behind it properly.

Answer

This comment is for research at the verification stage, not exploratory research. This study was based on epidemiology and statistics. These were self-explanatory and were not explained in the paper. We authored this paper for clinical researchers to help make new specific hypotheses. This study is an exploratory study to find a theoretical basis. Therefore, the rationale of the current study and the possible mechanism behind it does not yet exist and cannot be written.

The other oral fungal infection related patient information is not explored, such as oral hygiene, diets, diabetics and lifestyles, etc.

Answer

These data could not be obtained from the medical records. When there is no data on oral hygiene, diets, diabetics and lifestyles, etc., there is no method to explore them. It is a mistake to demand an explanation for something that could not be done due to research limitations. When there is no patient information, such as oral hygiene, diets, diabetics and lifestyles, etc., if you know the method to explore them, please tell us it.

We have worked hard to incorporate your feedback and hope that these revisions persuade you to accept our submission.

Reviewer 3 Report

Comments and Suggestions for Authors

Dear authors,

The manuscript has been improved. It may be processed further.

Comments on the Quality of English Language

Although the authors have corrected English, it may be improved during proofreading.

Author Response

Dear Reviewer 3,

The manuscript has been improved. It may be processed further.

Thank you very much for reviewing our manuscript and offering valuable advice.

This study only sheds light on some of the ambiguity. We will proceed with this research. Again, thank you for reading the ambiguous and difficult-to-read text and pointing out the appropriate points.

Reviewer 4 Report

Comments and Suggestions for Authors

The authors significantly improve the manuscript's quality. However, some modifications need to be made.

 Abstract:

Restructure the abstract in a more organized and logical manner, emphasizing the main study results. Avoid starting sentences with numbers. Include "A total of" before 314 patients.

Introduction:

Several passages lack appropriate references. Authors should take care to cite the relevant literature properly.

Lines 34 to 36 - The authors quote, "However, many of these papers reported analytical results that divided continuous data into arbitrary segments that reduced the amount of information contained in the continuous data." Please provide the appropriate citation here and rephrase this sentence for better clarity. It is confusing.

Materials and Methods:

Provide a succinct description of how mass spectroscopy using a MALDI biotype was conducted. This may include details about the procedures, specific equipment used, and any special methods applied.

Discussion:

The discussion requires significant improvement. Once again, several sentences lack proper citations.

Explain how the results align or differ from the existing literature in the field. Highlight the practical significance of the results and their potential impact on clinical practice, health policies, or future research.

Dedicate a paragraph to discuss the study limitations objectively and in detail. Identify potential methodological biases and briefly explain their impact on the results. Conclude by proposing additional research areas or follow-up studies to address pending or emerging questions.

Author Response

Dear Reviewer 4,

Thank you very much for reviewing our manuscript and offering valuable advice.

We have addressed your comments with point-by-point responses, and revised the manuscript accordingly. To make our present study easier to understand, we modified our manuscript according to your suggestion.

Abstract:

Restructure the abstract in a more organized and logical manner, emphasizing the main study results. Avoid starting sentences with numbers. Include "A total of" before 314 patients.

Answer

Thank you for your kind suggestion. We modified our manuscript. The structure is logical, but to help understand We added statistical interpretation in discussion. This is because there are various schools of statistics.

Introduction:

Several passages lack appropriate references. Authors should take care to cite the relevant literature properly.

Lines 34 to 36 - The authors quote, "However, many of these papers reported analytical results that divided continuous data into arbitrary segments that reduced the amount of information contained in the continuous data." Please provide the appropriate citation here and rephrase this sentence for better clarity. It is confusing.

Answer

Thank you so much for your suggestion. These were ambiguous and difficult-to-read. We decided to explain this with a concrete example. This is an important point to make in emphasizing the benefits of this paper.

We can provide another example.

X1 : = (Marked improvement, Moderate improvement, Mild improvement, No change, Worsening) (Vector representation of variables representing order.)

X2 : = (8, 12, 45, 35, 0) (A vector representing observations for Group 1.)

X3 : = (0, 25, 25, 35, 15) (A vector representing observations for Group 2.)

These vectors represent ordinal data. When performing a U-test on the overall degree of improvement, Group 1 showed better improvement (p = 0.04). In this case, the results are reliable.

By setting a cutoff point based on a certain ranking, this data can be segmented.

When classifying mild improvement or better as an improvement, the improvement rate for Group 1 is 65% and that for Group 2 is 50% (chi-square test p value is p = 0.0452). Similarly, when a moderate or higher improvement is defined as an improvement, the improvement rates are reversed between Groups 1 and 2, with Group 1 having a 20% improvement rate and Group 2 having a 25% improvement rate. These results are less reliable and rougher.

It is our belief that comparisons to research results that show such trends should not be made.

The references were checked and then rewritten.

Materials and Methods:

Provide a succinct description of how mass spectroscopy using a MALDI biotype was conducted. This may include details about the procedures, specific equipment used, and any special methods applied.

Answer

Thank you for your kind suggestion. It was necessary to describe how we used the equipment.

Discussion:

Answer

Thank you for your kind suggestion.

The discussion requires significant improvement. Once again, several sentences lack proper citations.

Answer

The references were checked and then rewritten.

Explain how the results align or differ from the existing literature in the field. Highlight the practical significance of the results and their potential impact on clinical practice, health policies, or future research.

Answer

Comparisons using the same scale are preferable. However, even if we don't compare in detail, the results were generally consistent with existing rough results. Since the comparison data retrieved from the literature are at the lower of the scale, a good comparison cannot be made. There is no point in comparing the ratio scale (accurate age) with extreme ages such as those over 65 or 80 years old (nominal scale or ordinal scale). Moreover, the study we referred to is a preliminary study in which Candida identification is based on colony color. Rather than comparing the results, it is more profitable to accurately estimate the true values using interval estimations, as we did.

Dedicate a paragraph to discuss the study limitations objectively and in detail. Identify potential methodological biases and briefly explain their impact on the results. Conclude by proposing additional research areas or follow-up studies to address pending or emerging questions.

Answer

By summarizing the results, we made clear the strong and weak points of this paper. Accurate prevalence surveys were conducted, and various variables were summarized in tables for use in meta-analyses. Added sample size calculations and briefly described meta-analysis methods.

Again, thank you for giving us the opportunity to strengthen our manuscript with your valuable comments. Our manuscript has been reorganized based on your kind and academic comments. We have worked hard to incorporate your feedback and hope that these revisions persuade you to accept our submission.

Round 2

Reviewer 1 Report

Comments and Suggestions for Authors

The A.A. have addressed all the reviewer's suggestion.

Reviewer 2 Report

Comments and Suggestions for Authors

OK. I accepted your argument of the study design about the new hypothesis and research limitations and could not measure the other confounding factors.